# Predicting Intoxication Using Motorcycle and Head Movements of Riders Wearing Alcohol Intoxication Goggles

**Rosemary Seva [1,*], Imanuel Luir del Rosario [1], Lorenzo Miguel Peñafiel [1], John Michael Young [1] and Edwin Sybingco [2]**

1    Department of Industrial and Systems Engineering, De La Salle University, 2401 Taft Ave., Malate, Manila 1004, Philippines; imanuel_delrosario@dlsu.edu.ph (I.L.d.R.); lorenzo_penafiel@dlsu.edu.ph (L.M.P.); john_michael_young@dlsu.edu.ph (J.M.Y.)

2    Department of Electronics and Computer Engineering, De La Salle University, 2401 Taft Ave., Malate, Manila 1004, Philippines; edwin.sybingco@dlsu.edu.ph

\*    Correspondence: rosemary.seva@dlsu.edu.ph

**Abstract:** The movement of a motorcycle is one of the critical factors that influences the stability of the ride. It has been established that the gait patterns of drunk and sober people are distinct. However, drunk motorcycle (MC) drivers' balance has not been investigated as a predictor of intoxication. This paper characterized and used MC and head movements, such as pitch and roll, to predict intoxication while riding. Two separate experiments were conducted to monitor MC and head movement. Male participants were recruited between the ages of 23 and 50 to participate in the study. Participants used alcohol intoxication goggles (AIG) to simulate blood alcohol content (BAC) while driving on a straight path. Placebo goggles were used for control. Results showed that pitch and roll amplitudes of the MC could distinguish drivers wearing placebo and AIGs, as well as the pitch and roll frequency of the head. Deep learning can be used to predict the intoxication of MC riders. The predictive accuracy of the algorithm shows a viable opportunity for the use of movement to monitor drunk riders on the road.

**Keywords:** accident prevention; alcohol intoxication; deep learning

## 1. Introduction

Drunk motorcycle (MC) drivers are more likely to be responsible for fatal single-vehicle crashes compared to drunk car drivers [1] and have an increased injury severity [2–4]. Drunk driving is highly associated with increased injury severity and head injuries among MC drivers because their protective gears may not be enough to cushion the impact of a collision [5]. Severe injuries among MC riders were found to be linearly related to the blood alcohol concentration (BAC) of the drunk drivers [2]. Injury-related data in the Philippines from 2010–2019 showed that MC drivers are prone to fatal injuries and alcohol use is related to multiple injuries [6].

The use of sobriety checkpoints proved to be an effective and cost-efficient way of preventing drunk driving [7]. One way of determining a person's alcohol level without a breathalyzer is the Standardized Field Sobriety Test (SFST) comprised of three tests performed outside of a vehicle: the horizontal gaze nystagmus (HGN), the walk-and-turn, and the one-leg stand tests [8]. The HGN test is related to tracking eye movement, while the walk-and-turn and one-leg stand tests are both related to the ability to balance. Based on the nature of these tests, attention and balance are crucial aspects of sobriety. In some MC training schools, a person's readiness to drive an MC is assessed by balancing a two-wheeler while traversing a figure eight track.

The movement of a drunk person is distinct from a sober one. For standing subjects, intoxication causes greater body sway [9], deviation from body alignment [10], increased movement spans of the head, shoulders, hip, and knees in anteroposterior and lateral

directions [10], and balance impairment [11]. Rudin-Brown et al. [11] found that a person's lateral and anterior-posterior sway using XYZ axes differ between sober and drunk persons, making this information usable in prediction. Nassi et al. [12] used the gait pattern of walking subjects, using sensors from wearable devices to predict alcohol intoxication.

Even the riding patterns of drunk and sober drivers are different. Creaser et al. [13] found that BAC is inversely related to the riding skill of MC drivers. There is a significant difference in the riding pattern of sober and drunk riders in terms of errors and response times. BAC levels above 0.05% causes a complex, multifaceted deterioration of human postural control that compromises the speed of movement and balance of the rider [14]. Even a low dose of alcohol lowers the total reaction times for visual and audio stimuli [15].

Previous investigations on the effect of alcohol on movement have been done on intoxicated persons that are statically standing or with minimal body movement. There are no known methods of detecting whether an MC rider is intoxicated while riding the MC. Hafström, Patel, et al. [16] showed that poor visual input caused by intoxication and disrupted vestibular information cause unreliable posture feedback. The head loses its vertical position and deviates to the left. Modig et al. [14] suggested that high BAC levels resulted in a person's inability to control balance, resulting in falls. This can also happen to a person riding an MC.

This study aims to show that the MC and head movements of drunk and sober MC riders are distinct. It will integrate previous research findings on the relationship between intoxication and movement in the context of MC driving. The authors are unaware of any investigation relating MC balance and head movement to intoxication while riding an MC.

It is hypothesized that the MC and head movement can be used to predict intoxication. As Hafström, Patel, et al. [16] discovered, an increased BAC level significantly affected head position due to decreased visual input. The head is more anterior as visual input is decreased.

## 2. Research Method

Two experiments were conducted to investigate MC and head movements. The MC movement experiment was conducted months ahead of the head movement experiment. Thus, lessons learned from the MC movement experiment were implemented in the head experiment. The two experiments differed in terms of the number of participants and the independent variables investigated.

### 2.1. Experiment 1: MC Movement

2.1.1. Participants

Twenty (20) healthy male participants (age: 26.1 ± 2.86 years) were recruited for this experiment. They are licensed MC drivers with at least 5 years of MC driving experience and are legally allowed to consume alcohol. The study design was approved by a panel of examiners. A written waiver was obtained from all participants before the experiment.

2.1.2. Experimental Set-Up

The MC type used was the Honda Wave 110 MC (Honda, Tokyo, Japan), which is very common in the Philippines. All participants were asked to wear personal protective equipment (PPE), such as a helmet, a pair of riding gloves, elbow pads, knee pads, and shin guards. They were asked to bring their own above-ankle boots during the experiment.

The experiment was performed on a 55 m-long road without any obstacles, as can be seen in Figure 1. The course is meant to simulate the first few meters of a drive because the device should be able to monitor the condition of the driver at the initial stage of the ride.

In this experiment, the movement of the motorcycle was measured, which we called "MC movement pattern". The pattern of the MC movement was measured using the Inertial Measurement Unit (IMU) device Genuino 101. It is classified as a learning and development board that runs on the Intel Curie Module and can process real-time data useful in monitoring movement patterns. It is a programmable device that captures and

processes the X (roll), Y (pitch), and Z (yaw) movements in real-time with a 3D model to simulate the motion. As can be seen in Figure 2, pitch refers to movements in the posterior-anterior (PA) plane, while roll refers to movement in the medial-lateral (ML) plane. Since the data gathered from accelerometers are analog, the device was programmed to automatically translate the XYZ movement readings into digital data. Although the device collected XYZ data, only the XY data were used in this study.

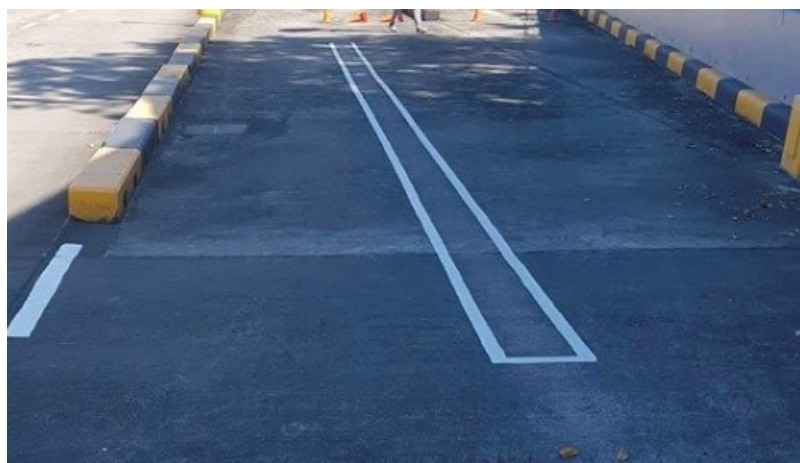

**Figure 1.** Rider's track.

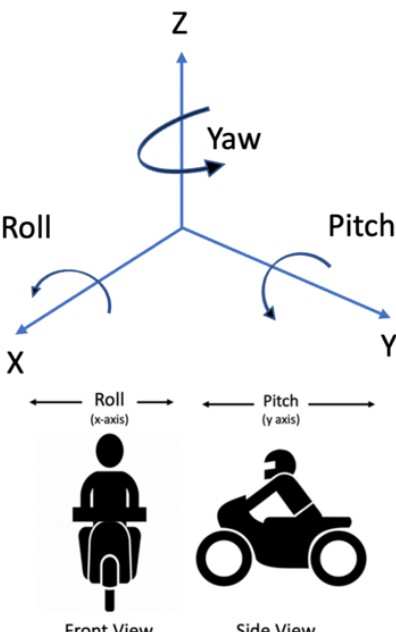

**Figure 2.** Motorcycle movement axes.

The IMU was mounted at the rear of the driver's seat. Data recording was initiated at the start of the course and ended at the designated end of the course. The sampling rate of the sensor was 50 Hz.

Due to obvious hazards of actual intoxication, alcohol intoxication goggles (AIGs) were used to effect alcohol intoxication among participants. AIGs distort images to simulate the effect of alcohol without affecting cognitive processes. AIGs have been used as an alternative approach to replicate alcohol-related vision impairment or the impact of alcohol intoxication on simulated driving performances, especially during driver education programs [17]. These goggles can simulate different levels of BAC. During the experiment, participants were required to use their prescription glasses to guarantee that vision impair-

ment would solely come from using AIGs. Each participant was fitted with an appropriate goggle prior to wearing a full-face helmet provided by the research team. It was ensured that the goggles were worn properly with the helmet to avoid errors.

### 2.1.3. Experimental Design and Procedure

The dependent variables are the pitch and roll characteristics of the MC. The peak-to-peak amplitude of both roll and pitch were considered because they show the range of movement of the MC, while frequency indicates the number of times the MC crossed the body plane.

Pitch frequency (PF) and roll frequency (RF) were measured by counting the number of times the MC crossed the frontal and lateral planes, respectively. Pitch amplitude (PA) and roll amplitude (RA) were measured by the angle of deviation of the MC from the frontal and lateral plane, respectively. In this experiment, it is expected that a drunk driver is more likely to have a problem balancing the MC that can be captured by roll data. Pitch was also used as a predictor because the inability of the drunk driver to accurately perceive the surroundings will cause constant braking and variation in speed. It was measured by counting the number of times the MC accelerated and decelerated. MC roll was measured by counting the number of times the MC banked left and right.

There is only one independent variable in this experiment: the level of intoxication measured as BAC level.

Each participant performed three driving tasks using three different goggles: (1) placebo goggles, (2) AIG estimated BAC 0.050%, and (3) AIG estimated BAC 0.080%. All participants completed all trials in one visit. The conditions administered to participants were randomized to address learning bias. For the remaining part of this manuscript, the words sober and drunk will be used to mean the use of placebo goggles and AIGs, respectively.

After the screening, qualified participants were informed about the procedure of the experiment. A dry run was conducted to familiarize the MC riders with the course, the vehicle, the AIG, and the measurement system to be used.

For each trial, the rider was asked to wear the appropriate AIG and traverse the course from a designated beginning and end. Data acquisition was triggered at the beginning of the ride and stopped when the rider reached the end of the course. Data were compiled manually after each trial.

### 2.2. Experiment 2: Head Movement

#### 2.2.1. Participants

Fifteen healthy male participants (age: 27 ± 4.25 years) were recruited for the head movement experiment with the same profile as the MC movement experiment.

#### 2.2.2. Experimental Set-Up

The same type of MC was used for the experiment: the Honda Wave 110 MC. All participants were asked to wear the same PPEs as in the first experiment and their own above-ankle boots during the experiment.

A longer course was traversed by the participants in this experiment, which is 100 m without any obstacles. The riding path was extended to have more observations of head movement.

The same apparatus was used for the head experiment except it was mounted on top of the helmet worn by the rider. Data recording was initiated at the start of the course and ended at the designated end of the course. The sampling rate of the sensor was reduced to 20 Hz to compensate for the increase in the distance travelled in this experiment.

AIGs were used to produce the effect of intoxication among participants for the reasons explained earlier. Participants were also required to use their prescription glasses to isolate the effect of AIG on vision. It was ensured that the goggles were worn properly by the participants with the helmet before commencing the experiment.

### 2.2.3. Experimental Design and Procedure

The dependent variables for this experiment are pitch and roll of the head. Head pitch was tracked because a drunk person's head, while standing, tends to be more anterior [10]. Since the MC rider's head position is dynamic, it is hypothesized that the tendency of the head to drop while drunk is constantly corrected by the rider throughout the course, so the pitch frequency and amplitude are expected to be higher for drunk riders. Head pitch frequency, also called body sway, has been used to measure balance in dynamic posturography studies of drunk individuals [18]. On the other hand, lateral head movement (roll) was measured because it is related to balancing the MC by bending the head from left to right. Since the vision of drunk drivers is impaired, balancing the vehicle can be a challenge, so the roll frequency and amplitude will also be higher.

The drivers performed two driving tasks with repetition: (1) placebo goggles and (2) AIG estimated BAC 0.050%. Higher BAC levels were not considered for the head movement experiment because initial findings from the MC movement study showed that the 0.08% BAC level severely affected vision. Moreover, the legal alcohol limit in the Philippines, based on the Republic Act 10,586 of 2013, is 0.05% BAC, so any change in movement should be observable at this level. All participants completed all trials in one visit. The conditions administered to participants were randomized to address the learning effect.

### 2.3. Statistical Analysis

Data characteristics were analyzed using descriptive statistics. Differences between treatments were tested using ANOVA, and a paired t-test was used for the MC and head movement experiments, respectively, at a significance level of 0.05.

Data collected from motion and inertial sensors, such as accelerometer and gyroscope, have been applied to Long short-term memory (LSTM)-based deep neural network to predict human physical activities, such as the papers proposed by [19,20]. In this study, data from the two experiments were used for prediction using a deep learning algorithm. The LSTM extracted the features, then the features were applied to a fully connected layer to perform the prediction.

### 2.3.1. Classifier

The classification network architecture used in this study is shown in Figure 3. Three types of input sequence are considered in the deep LSTM classifier. The first model considered is the time frequency representation of roll signal, the second model is the time frequency representation of pitch signal, and the third model is the combination of roll and pitch. The LSTM is configured with 50 hidden units. The output of the last time step of the sequence output is connected to the fully connected layer. The fully connected layer has two outputs corresponding to the drunk and not drunk classes. The softmax uses a softmax function to ensure that the output is in the range of 0 to 1. The classification will compute cross entropy, which will produce the corresponding classification.

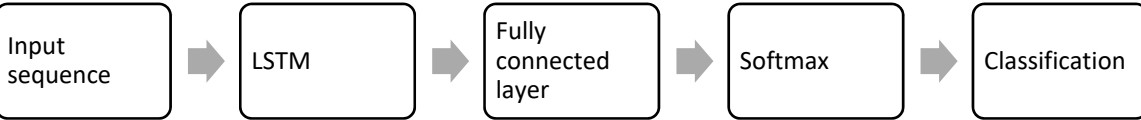

**Figure 3.** Deep LSTM Classifier.

Shown in Table 1 are the details of the different layers or stages of the deep LSTM classifier in terms of the number of activations per layer and the corresponding learnable parameters. The first stage will transform the pitch/roll sequence into a time–time frequency sequence with 32 frequency features. The output of the input sequence is applied to the second stage, which is the LSTM. The LSTM will extract the time–frequency features

into 50 features which are then applied to the classifier (stages 3 to 5) to predict the input sequence as "Drunk" or "Not Drunk".

**Table 1.** Details of the different layers of the Deep LSTM Classifier.

| Stage | Type | Activation | Learnable Parameters | |
|---|---|---|---|---|
| 1 | Input Sequence | 32 | | |
| 2 | LSTM | 50 | Input Weights | $200 \times 32$ |
| | | | Recurrent Weights | $200 \times 50$ |
| | | | Bias | $200 \times 1$ |
| 3 | Fully Connected | 2 | Weights | $2 \times 50$ |
| | | | Bias | $2 \times 1$ |
| 4 | Softmax | 2 | | |
| 5 | Classification Cross entropy with class "Drunk" and "Not Drunk" | | | |

### 2.3.2. Input Sequence Representation for MC Movement Experiment

The input sequence was generated from the time–frequency representation of the roll and pitch signals of the MC movement. In this experiment, 40 trials were classified as "Drunk" [0.05% BAC and 0.08% BAC] and 20 trials as "Not Drunk". For each trial, the roll and pitch signals were recorded.

The number of samples of the roll and pitch signal varies from 529 to 1268. Shown in Figure 4 is the distribution of the number of frames for the different trials. For each set of data, the time–frequency representation of the roll and pitch is computed by windowing using a Hanning window with 32 samples and 8 sample overlaps, resulting in 32 frequency samples by $n$ number of frames, where $n$ varies from 21 to 52.

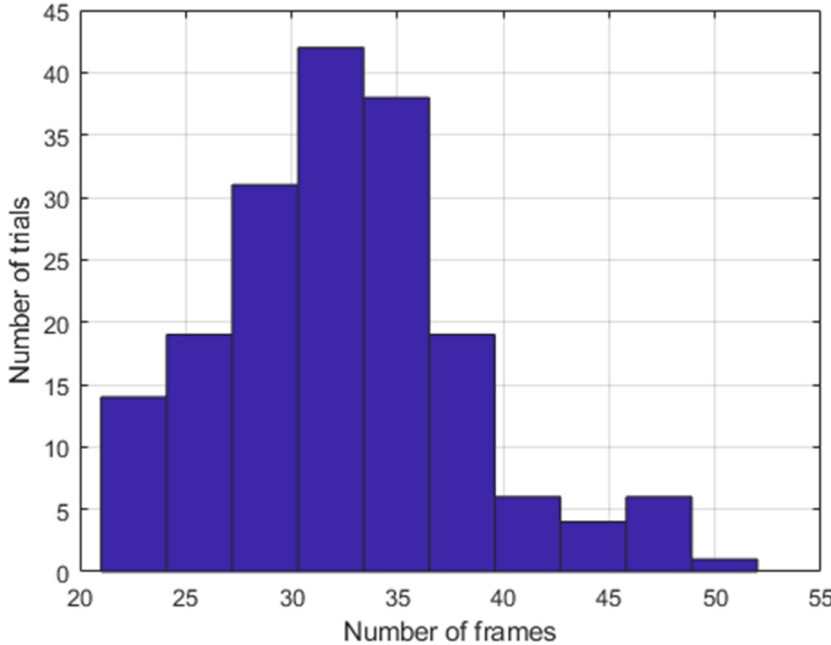

**Figure 4.** Number of frames histogram (MC movement experiment).

Shown in Figure 5 is the time series calculation of the input sequence at time step t. This is denoted by $x_t$, with 32 features and n number of frames or time steps applied through the long short-term memory (LSTM) blocks. The hidden state, which is also the

output state, is denoted by $h_t$ and the cell state is denoted by $c_t$ at time step t. The hidden state is a vector with a length equal to the number of hidden units.

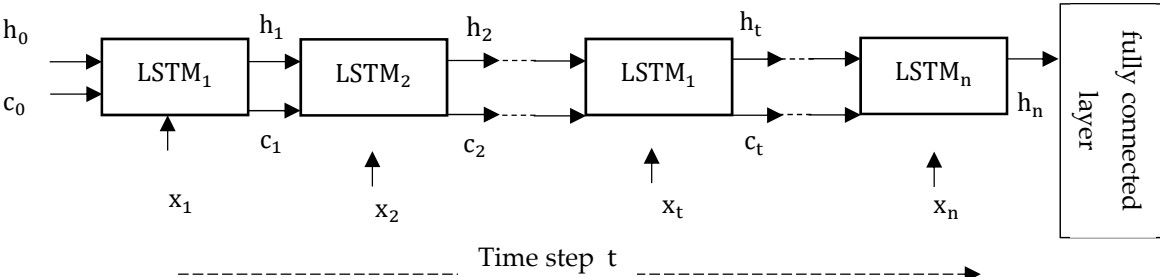

**Figure 5.** LSTM layer architecture.

### 2.3.3. Input Sequence Representation for Head Movement Experiment

The input sequence was generated from the time–frequency representation of the roll and pitch signals of the head movement. For the head movement experiment, 30 trials were classified as "Drunk" and 30 trials as "Not Drunk." The number of samples of the roll and pitch signal varied from 544 to 1117. The time–frequency representation of the roll and pitch was computed by windowing using a Hanning window with 32 samples and 8 sample overlaps, resulting in 32 frequency samples by $n$ number of frames. From the dataset, the number of frames, $n$, varied from 5 to 9. Shown in Figure 6 is the histogram of the number of frames for all the trials.

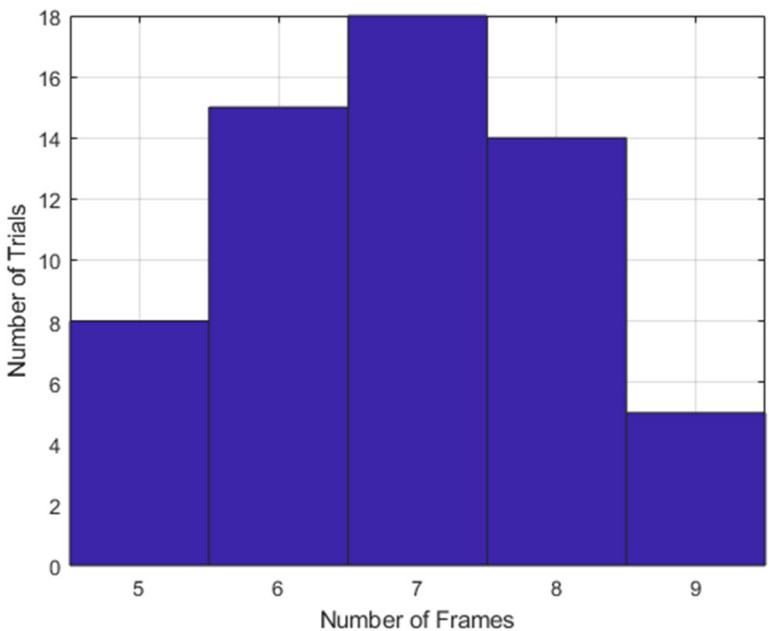

**Figure 6.** Number of frames histogram (head movement experiment).

### 2.3.4. LSTM Block and Training Configuration and Performance

Each LSTM block [21] used the current state of the network represented by $(c_{t-1}, h_{t-1})$ and the input sequence $x_t$ to compute the output state, $h_t$, and update the cell state, $c_t$. The cell state and the hidden state at time step t were computed using the following equations:

$$c_t = f_t \odot c_{t-1} + i_t \odot g_t \tag{1}$$

$$h_t = o_t \odot \sigma_c(c_t) \tag{2}$$

where $\odot$ is a Hadamard product, $\sigma_c$ is the state activation function that uses the hyperbolic tangent function, $o_t$ is the output gate, $g_t$ is the cell candidate, $i_t$ is the input gate, and

$f_t$ is the forget gate. Each element of the LSTM blocks has a learnable weight defined as the input weights, $W_k$, recurrent weights, $R_k$, and bias, $b_k$. The index k corresponds to i for input gate, f for forget gate, g for cell candidate, and o for output gate. The following formulas were used to compute the following elements of the LSTM blocks:

$$o_t = \sigma(W_O x_t + R_O h_{t-1} + b_O) \tag{3}$$

$$g_t = \sigma_c(W_g x_t + R_g h_{t-1} + b_g) \tag{4}$$

$$f_t = \sigma(W_f x_t + R_f h_{t-1} + b_f) \tag{5}$$

$$i_t = \sigma(W_i x_t + R_i h_{t-1} + b_i) \tag{6}$$

where $\sigma$ is the gate activation function that uses the sigmoid function.

The deep LSTM network was trained using an Adam optimizer. To prevent gradient explosion, the gradient was clipped with a threshold of one. The training data was sorted from lowest to highest and divided into a minibatch of 10 for MC movement and 5 for head movement to minimize the effect of padding. The algorithm padded the sequences such that the sequence had the same length as the longest sequence to ensure that each minibatch had the same size. The dataset was split into two to test the ability of the deep LSTM network to generalize from half of the dataset. The first half was used to train the deep LSTM network and the other half was used for testing.

All data were analyzed in post-processing by using the MATLAB release 2022a by The Mathworks Inc, Natick, MA, USA [22].

## 3. Results

### 3.1. MC Movement Characteristics

Descriptive statistics on MC movement are shown in Table 2. Pitch and roll frequencies have high standard deviation values, indicating differences in participants' balancing effort (roll) and braking pattern (pitch). There is no significant pitch and roll frequency increase between 0% and 0.05% BAC. The pitch amplitudes decrease while the roll amplitudes increase with the BAC level. It can also be observed that frequency and amplitude values for pitch and roll are inversely related.

**Table 2.** MC movement parameters, comparison between BAC level (degrees).

| | Mean $\pm$ SD | | |
|---|---|---|---|
| **Variable\BAC Level** | **0% BAC** | **0.05% BAC** | **0.08% BAC** |
| Pitch frequency | $11.0 \pm 8.5$ | $12.2 \pm 11.265$ | $16.2 \pm 8.7$ |
| Roll frequency | $30.0 \pm 9.0$ | $28.2 \pm 11.004$ | $24.8 \pm 9.0$ |
| Pitch amplitude (deg) | $1.2 \pm 0.4$ | $0.8 \pm 0.3$ | $0.6 \pm 0.2$ |
| Roll amplitude (deg) | $0.14 \pm 0.1$ | $0.2 \pm 0.097$ | $0.2 \pm 0.1$ |

One-way ANOVA was performed to compare the effect of the AIG BAC levels to the frequency and amplitude of pitch and roll. Results showed a significant effect of BAC levels to pitch amplitude (Wilk's Lambda = 0.281, F = 22.996, $p < 0.001$) and roll amplitude (Wilk's Lambda = 0.535, F = 7.827, $p < 0.01$). Post-hoc comparisons using the Bonferonni test showed that the mean pitch amplitudes of the three conditions are significantly different. For roll amplitude, the mean roll amplitude of BAC levels 0.05 and 0.08 were significantly different from 0. However, there was no significant difference in the mean roll amplitude between BAC levels 0.05 and 0.08.

### 3.2. Head Movement Characteristics

Head movement characteristics are summarized in Table 3. The frequency of changes in head pitch ($p = 0.003$, paired *t*-test) and roll ($p = 0.033$, paired *t*-test) of drunk riders were significantly higher than sober ones. On the other hand, the pitch and roll amplitudes of

both riders have no significant difference. Frequency and amplitude of the head are both higher for pitch compared to roll.

**Table 3.** Head movement parameters, comparison between BAC levels (degrees).

| | Mean ± SD | | |
|---|---|---|---|
| **Variable** | **0% BAC** | **0.05% BAC** | **_p_-Value** |
| Pitch frequency | 50.7 ± 11.5 | 57.3 ± 14.5 | 0.00 |
| Roll frequency | 50.0 ± 10.7 | 54.6 ± 13.0 | 0.03 |
| Pitch amplitude (deg) | 4.0 ± 1.0 | 3.7 ± 0.9 | 0.10 |
| Roll amplitude (deg) | 1.7 ± 0.9 | 1.7 ± 0.8 | 0.40 |

*3.3. Deep LSTM Training Configuration and Performance*

3.3.1. MC Movement Experiment

Results obtained using LSTM were used to predict a rider's state of intoxication using pitch frequency and amplitude data as can be seen in Table 4. The training data had an accuracy of 100% for the three models, while the test data had a lower accuracy of 43–77%. Three models were evaluated: roll only, pitch only, and roll-pitch data. The roll-pitch is the best model considering its overall performance. The high precision value indicates the model's ability to identify intoxicated riders.

**Table 4.** Performance metrics of various approaches (MC experiment).

| Model | Confusion Matrix | | Accuracy | Precision | Recall | F1 |
|---|---|---|---|---|---|---|
| Roll Only | 20<br>8 | 0<br>2 | 72% | 100% | 71% | 83% |
| Pitch Only | 4<br>1 | 16<br>9 | 43% | 20% | 80% | 32% |
| Roll-Pitch | 19<br>6 | 1<br>4 | 77% | 95% | 76% | 84% |

3.3.2. Head Movement Experiment

Table 5 shows the performance metrics of the deep learning model in the head movement experiment. The accuracy of test data ranged from 37–67%. Of the three models considered, the pitch only model has the best performance for all the metrics considered. The precision value of 93% suggests the model's capability to distinguish drunk riders.

**Table 5.** Performance metrics of various approaches (head experiment).

| Model | Confusion Matrix | | Accuracy | Precision | Recall | F1 |
|---|---|---|---|---|---|---|
| Roll Only | 4<br>8 | 11<br>7 | 37% | 27% | 33% | 30% |
| Pitch Only | 14<br>9 | 1<br>6 | 67% | 93% | 61% | 74% |
| Roll-Pitch | 6<br>3 | 9<br>12 | 60% | 40% | 67% | 50% |

## 4. Discussion

*4.1. MC Movement Experiment*

Movement dynamics of the MC is greatly affected by its rider [23]. Results validated the hypothesis that MC movement of sober and drunk riders are distinct. All dependent variables had a significant difference at 0.08% BAC level, indicating that visual impairment caused by intoxication affects movement and balance. At 0.05% BAC level, however, only the pitch and roll amplitude had a significant difference between sober and drunk

riders. The high roll amplitude can be caused by the difficulty in balancing the MC, resulting in larger movement displacement taking more time to correct. This explains the negative relationship between the frequency and amplitude values in Table 2. Riders had to constantly change the sagittal position of their MC and rely on proprioception to balance.

The results affirm the importance of visual feedback for balancing. Dynamic balance was shown to be affected by impaired vision, manifested by greater sway in the ML plane [24]. Poor visual information deteriorates postural stability, resulting in greater body sway in people with low vision compared to normal vision in dynamic movement tests, such as walking in tandem [25]. Modig, Fransson, Magnusson, and Patel [14] discovered that BAC levels greater than the legal limit of 0.05% cause the body to be unstable due to deterioration in posture, especially in the lateral direction, and this was also evident even at BAC levels as low as 0.03% [18]. Vision dominates the information coming from the inner ear and the receptors in the muscles and joints [26]. There are no studies involving MC movement while riding, but it is apparent that the effect of poor vision on MC balance is the same.

The riders' driving behavior also changed as the BAC level increased. The difference between the pitch frequency and amplitude between drunk and sober riders can be explained by the cautiousness of drivers when their vision is impaired by the AIGs. They travel at a slower rate, braking more frequently with a force that is not large enough to create a great sway in the PA plane. Thus, at higher BAC levels, the sway of the MC in the PA plane decreases while the frequency increases.

The difference in the movement patterns of drunk and sober drivers prompted the modeling of the relationship between movement and BAC level. Deep learning was used because of its ability to extract high level features from data [27].

Both roll and pitch data were identified as important features using a deep learning algorithm to classify sober and drunk riders. The combined features can predict drunk riders 95% of the time. However, the lower accuracy indicates the bias for predicting drunk compared to sober riders. Accuracy was lower compared to precision because more data were collected from drunk riders.

### 4.2. Head Movement Experiment

The amplitudes of head movement of both sober and drunk riders have no significant difference. Roll and pitch frequency were significantly higher with the use of 0.05% AIG, which is reasonably caused by poor visual information. Timmis and Pardhan [28] found that visually impaired subjects were found to have higher lateral head movement and movement corrections compared to normal vision participants when completing a visually guided, manual prehension task. Low vision subjects tend to have fast vertical nystagmus (up-down movement of the eyes) that was correlated with vertical head movement [29]. For MC riders, this could have been an adaptive behavior to find a head position that would improve environmental perception. The head is moved frequently in the vertical and lateral direction to correct the vision that was limited by the 0.05% AIG.

Pitch was found to be the most important feature that could be used to classify sober and drunk riders using head movement. The 93% precision for the pitch only model is significantly higher than the roll only and pitch-roll combination models. The accuracy is relatively low but can still be improved with additional data. The higher precision value compared to accuracy indicates a bias in identifying drunk riders. The relevance of pitch in the model validates the hypothesis that the rider conducts posture correction at the PA plane because of the tendency of the head to drop.

This study validated that movement can be used to predict intoxication among MC riders. The predictive accuracy of the algorithm shows a viable opportunity in the use of movement as a way of monitoring drunk riders on the road, especially in densely populated cities where it is impractical to apprehend suspected drunk MC drivers. The movement of MC riders can be monitored via a smart system that can provide information to authorities. For example, sensors that monitor erratic movement that can signify drunkenness or

the inability to balance can be integrated by manufacturers to the MCs as an additional safety feature. The sensor can provide the rider or the authorities with a signal that the rider is not able to balance the MC well due to intoxication. This trend has been apparent in automakers that integrate safety features of alcohol intoxication monitoring using breathalyzers and touch sensors [30]. Smart helmets have already been designed to monitor alcohol intoxication among MC riders with the use of an alcohol sensor [31], but the relationship of movement to intoxication has not been explored. This study provides an alternative method for monitoring intoxication unobtrusively. The International Motorcycle Manufacturers' Association (IMMA) is a federation of regional and national associations that seeks to promote solutions for safe MC mobility [32]. The knowledge obtained from this research can be integrated into manufacturing safer two-wheeled vehicles. Monitoring of MC movement can be integrated into the MC design while the head movement can be part of the helmet design. These two safety variables can provide information about the driver's performance and allow for early identification of balance issues that can result from intoxication.

Alternatively, this method can be used in licensing MC riders. The ability to balance the MC is an important skill for new riders. The movement monitoring system can be used to gauge the readiness of the rider to handle an MC and get a license.

There is a potential for law enforcement agencies to use this technology for contactless apprehension of drunk MC drivers if the accuracy can be improved. Governments can influence manufacturers to integrate the feature of movement monitoring in the product development stage of MCs. Policies can be developed to empower manufacturers to make smarter MCs that can detect drunk driving and notify surrounding people or drivers of the driver's condition.

This study has several limitations, foremost of which is the use of the AIG that only simulated the visual effects of alcohol and not the behavioral effects. Alcohol does not only affect vision, but also reaction time and motor skills that are also important for driver safety. However, due to the safety implications of having the drivers consume alcohol for this experiment, only the vision impairment was simulated. Future studies can consider a safer way of conducting experiments with actual drunk MC drivers to determine differences in the patterns of movements. Female drivers were also not represented in the sample. It is possible that the movement characteristics of female drivers are also distinct from male drivers. Finally, the number of data points for sober could have been increased to prevent bias in prediction.

## 5. Conclusions

Pitch and roll amplitudes of the body can reasonably distinguish drunk and sober drivers, as well as the pitch and roll frequency of the head. These movement characteristics have not been used in previous studies to detect intoxication. The results confirm earlier studies establishing the relationship between segmental body movement and alcohol level. Deep learning could be used to predict the intoxication of MC riders. The combination of pitch and roll data of the MC and head pitch are good predictors of MC rider intoxication. The model can be enhanced by conducting more validation experiments. The models obtained can be the basis for developing new movement monitoring technologies.

The model obtained from deep learning can be used to develop technologies that measure MC and head movement unobtrusively. Manufacturers can integrate these technologies when producing MCs or helmets in the future.

**Author Contributions:** Conceptualization, R.S., I.L.d.R., L.M.P. and J.M.Y.; methodology, R.S., I.L.d.R., L.M.P. and J.M.Y.; software, I.L.d.R., L.M.P., J.M.Y. and E.S.; validation, R.S., I.L.d.R., L.M.P. and J.M.Y.; formal analysis, I.L.d.R., L.M.P., J.M.Y. and E.S.; investigation, R.S., I.L.d.R., L.M.P., J.M.Y. and E.S.; resources, R.S.; data curation, I.L.d.R., L.M.P. and J.M.Y.; writing—R.S., I.L.d.R., L.M.P. and J.M.Y.; writing—R.S. and E.S.; visualization, E.S.; supervision, R.S.; project administration, R.S.; funding acquisition, R.S. All authors have read and agreed to the published version of the manuscript.

**Funding:** This research was funded by De La Salle University through its University Research and Coordination Office with Grant Number 57 F U 2TAY18-3TAY19 and the APC was funded by De La Salle University and the Engineering Research and Development for Technology.

**Institutional Review Board Statement:** The study was conducted in accordance with the Declaration of Helsinki, and the protocol was approved by the Ethics Committee of De La Salle University (protocol code DLSU-FRP.003.2018-2019.T2.GCOE approved 23 August 2019).

**Informed Consent Statement:** Informed consent/waiver was obtained from all subjects involved in the study.

**Data Availability Statement:** The data presented in this study are available on request from the corresponding author. The data are not publicly available due to data privacy constraints.

**Acknowledgments:** We acknowledge Arnica Joy Lacson, Javier Alfonso Prats, Gianella Supetran, and Carlos San Miguel for their help in conducting the experiments.

**Conflicts of Interest:** The authors declare no conflict of interest.

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
