# Peer review of "Predicting Intoxication Using Motorcycle and Head Movements of Riders Wearing Alcohol Intoxication Goggles"

_safety, 2023_

Round 1

Reviewer 1 Report (Previous Reviewer 2)

The issue taken up by the authors is important in terms of the safety of motorcyclists. However, it is worth making minor corrections in the manuscript.

1. The axis markings in Fig. 2 (axle diagram) are illegible. It's worth improving.

2. It might be worth considering how to break up the content in Chapter 2. It has a lot of very short sub-chapters.

3. The authors may extend the conclusion a bit

Author Response

Reviewer 2 Report (Previous Reviewer 3)

I would like to thanks the authors for all the work done to improve the presentation of their work.

Author Response

Thank you for your comment.

Reviewer 3 Report (New Reviewer)

Thank you for the opportunity to the review manuscript entitled: Predicting Intoxication using Motorcycle and Head Movements of Riders wearing Alcohol Intoxication Goggles

This is a very interesting study. I offer the following comments to the authors.

2.1.1 This section notes that participants had at least 5 years of driving experience. Is this driving a MC? Were they licensed to drive a MC? Needs clarification as has implications for bias in results if some were already MC licensed and some were not. And same for second experiment.

Did any participants do both experiments?  Important to clarify this.

Please explain how the goggles were appropriately fitted to someone wearing a helmet – was it a full face helmet? Was their distortion in how the goggles sat on the face due to appropriate helmet wearing?

Beginning of the Discussion notes that: All dependent variables had a significant difference at 0.08% BAC level indicating that visual impairment caused by intoxication affects movement and balance. However, the authors had previously stated that 0.08 goggles were not used for the second experiment. This requires clarification.

 The authors make some useful commentary in the Discussion about the potential to incorporate this kind of measurement into future manufacturing.

Appropriately written, only extremely minor errors which could be left unattended.

Round 2

Reviewer 1 Report (Previous Reviewer 2)

The paper after a minor correction of the authors can be published in Journal. 

This manuscript is a resubmission of an earlier submission. The following is a list of the peer review reports and author responses from that submission.

Round 1

Reviewer 1 Report

The paper brings up very important issue related to the riders’ safety. It is a good way to explore the effects of alcohol on motorcycle riders, since it would be too dangerous if they were really drunk. Although in this case there is no "feeling" of drunkenness in the subject's body, i.e. everything is focused on the fact that the sense of sight is little or significantly weakened when person in under influence, the experiments represent a good replica of the real situation and are well designed.

I have just two questions so maybe authors can answer it in the final version of the article:

Line 140 – could some other parameters influence the outcome? E.g. age, height, is the person usually drinking or drinking frequently etc.

Line 171 “The sampling rate of the sensor was reduced to 20Hz.” Why?

Reviewer 2 Report

The issue taken up by the authors is important in terms of the safety of motorcyclists. However, it is worth making significant corrections in the manuscript.

The method of citation is inconsistent with the requirements of the journal.

The use of subchapter 1.1 in the introduction seems completely redundant.

Line 58 "BAC levels 57 above 0.05%, the legal limit" varies from country to country. In some countries, this limit is 0. It is worth adding an additional comment

Chapter 2 describes in detail the research method used.

The description of the research method for 1 experiment is illegible. Too much text. After all, it is possible to present information in the text in the form of various diagrams or tables. This would greatly improve the comprehension of the text. This applies to, for example, apparatus and its installation, the test track, or Variables and measurements.

The number of subsections is too large. This also applies to section 2.2.

Any letter abbreviations should be explained in advance in the text

No explanation of what "*" means in table 1.

In Fig. 3, some full variable designations for values of h and x  are not visible. The figure should be corrected. Explanation of symbols from the drawing is required.

  Were the data presented in the table verified for normal distribution? The use of mean+/- SD can be used for such distributions. What are the reasons for such large SD values in relation to the mean value? In some cases, e.g. 0.05% BAC, they reach a value of over 90% of the mean value.

In the variables presented in the tables, it is worth indicating the units used.

It is worth making a brief commentary on the data presented in tables 2-5

It is worth extending the conclusion a bit, e.g. in terms of possible further actions.

Reviewer 3 Report

The title should mention the intoxication is simulated; otherwise, it leads to the impression that you evaluated drunk drivers, and the same applies to the rest of the manuscript. As an example, in the abstract, we cannot state that there are drunk and sober drivers. These are experimental conditions associated with the glasses. It is stated in the text at 2.1.5. Alcohol intoxication and thus should be stated from the beginning of the manuscript.

Line 27, the first sentence can be removed since the second sentence gives more details.

Line 56, mention what is the significant difference in riding ?

Line 80: how come a drunk detection system to a technology to train new drivers? Authors should be careful not going in too many directions with their introduction and focus solely on BAC detection.

a figure should've used to present the experimental setup and justify the YZ-only data

how can the body mvt can be recorded if the system is mounted on the driver's seat (line 114)

The reduction of the sampling rate from 50 to 20 should be justified.

2.2.6, why is Yaw not used in the head experiment ?

line 195 the BAC is 0,05 in ? It has to be specified where and a reference should be used. Why was 0,08 tested for the trunk (motorcycle mvt) then?

Why is the data compiled manually if you use a Genuino 101 that can record automatically the data?

2.3, an anova should be used for the trunk experiment and not a repetition of t-test. Moreover, 0,05 and 0,08 should be compared for sensibility of the tool developped.

2.4 section has to be back-up by references and the long-short term memory has to be described accordingly.

paragraph line 243: why have you chosen these parameters?

Figures' titles should be standardized

The number of digits after the dot should be standardized across data and tables as it varies from none to three.

line 348: where are these results coming from ?

Line 400: with the obtained results of accuracy, i would be more cautious with such a statement

I think the work is worthy of publication but requires considerable changes to the text to be more accurate in the term used and how the method is reported.

Reviewer 4 Report

The objective of the manuscript is to predict motorcycle rider’s intoxication using body and head movements. The topic of the manuscript is very interesting. The reviewer has a few comments.

1.    The literature review is incomplete. The authors may want to add more recent studies relevant to motorcycle rider’s intoxication over the world and countermeasures.

2.    The experiments appear to be very dangerous (even with the safety equipment). The reviewer wonders how the study design was approved by the Ethics Committee. What are the standards?

3.    Why only males were recruited for the two experiments? If the majority of motorcycle riders in Philippines are male, please provide the evidence.

4.    Provide more information about the AIG. How are their replicability of alcohol-related vision impairment. BAC levels affect riding ability. How were the BAC levels considered?

5.    The effects of alcohol intoxication is not only vision but also motor skills, reaction time, etc. Please discuss.

6.    Performance measures are not considered high. Are the models really capable to predict motorcycle rider’s intoxication?

7.    How can the developed model be useful? In the discussion, it was very briefly written. Include more policy implications.

Round 2

Reviewer 2 Report

The issue taken up by the authors is important in terms of the safety of motorcyclists. This version of the paper was modified.

The method of citation is still inconsistent with the requirements of the journal. In answer to the review, the author wrote,   "The initial submission to this journal does not require that the citation uses the requirement of the journal. The citation will be fixed after it is accepted for publication". According to this, in the finished version of the paper, the citations must be corrected.

Other comments have been taken into account

Author Response

The referencing style was changed to the requirement of the journal. 

Reviewer 3 Report

In their response to reviewer 4, the authors stated that the model is 100% accurate while all the information in Tables 4 & 5 shows an entirely different level of accuracy. Therefore, this should be clarified as it is not convincing as described now.

In their response to reviewer 3, the authors state, "The center of the body moves together with the motorcycle when trying to achieve balance. Since the system is mounted on the driver's seat, it will record the movement of the body core." This is inaccurate since one could move in the opposite direction of the motorcycle to counterbalance the center of mass. Therefore, one can not assume the driver's seat corresponds to the body core. A sensor on the driver seat represents the driver seat and should be presented as is.

Line 428, replace the can with could.

All the responses for reviewer 4 on the safety of the experiment should be added as limitations in the paper. While conducting such a controlled study, one has to assume that real-world data might differ significantly in terms of driving performance and of the accuracy of the model using IU.

Author Response

Comments

Reply

Reference to the new manuscript

In their response to reviewer 4, the authors stated that the model is 100% accurate while all the information in Tables 4 & 5 shows an entirely different level of accuracy. Therefore, this should be clarified as it is not convincing as described now.

The authors are referring to the training data that has an accuracy of 100%. The data presented in Tables 4 and 5 are for the test data and have lower accuracies.

In their response to reviewer 3, the authors state, "The center of the body moves together with the motorcycle when trying to achieve balance. Since the system is mounted on the driver's seat, it will record the movement of the body core." This is inaccurate since one could move in the opposite direction of the motorcycle to counterbalance the center of mass. Therefore, one can not assume the driver's seat corresponds to the body core. A sensor on the driver seat represents the driver seat and should be presented as is.

We called it the body movement experiment to differentiate it with the head movement experiment.

The body is composed of the upper and lower part. Counterbalancing the motorcycle is very evident while making sharp turns. It was observed that when making sharp turns, MC riders move their lower body. In the case of our experiment, the track is straight and relatively short. The one counterbalancing the weight is the upper part  of the body, the lower part of the body moves with the motorcycle. During the experiment, it was observed that the MC riders did not move their lower body.

Line 428, replace the can with could.

replaced can with could

Line 427

All the responses for reviewer 4 on the safety of the experiment should be added as limitations in the paper. While conducting such a controlled study, one has to assume that real-world data might differ significantly in terms of driving performance and of the accuracy of the model using IU.

They have been included in the limitations.

Lines 413-422

Reviewer 4 Report

The authors uploaded a wrong file of the responses. The "responses" is not the revised manuscript. So the authors are required to re-upload their response to reviewers' comments.

Author Response

REVIEWER 4

Comments

Reply

Reference to the new manuscript

The literature review is incomplete. The authors may want to add more recent studies relevant to motorcycle rider’s intoxication over the world and countermeasures.

Two recent articles were added to the review

Hamann, C. J., Wendt, L., Davis, J., Peek-Asa, C., Jansson, S., & Cavanaugh, J. E. (2022). Should we throw the book at 'em? Charge combinations and conviction rates among alcohol-influenced drivers involved in motorcycle crashes. J Safety Res, 83, 294-301. doi:10.1016/j.jsr.2022.09.003

Lin, H. A., Chan, C. W., Wiratama, B. S., Chen, P. L., Wang, M. H., Chao, C. J., . . . Pai, C. W. (2022). Evaluating the effect of drunk driving on fatal injuries among vulnerable road users in Taiwan: a population-based study. BMC Public Health, 22(1), 2059. doi:10.1186/s12889-022-14402-3

Lin, H. Y., Li, J. S., Pai, C. W., Chien, W. C., Huang, W. C., Hsu, C. W., . . . Lam, C. (2022). Environmental Factors Associated with Severe Motorcycle Crash Injury in University Neighborhoods: A Multicenter Study in Taiwan. Int J Environ Res Public Health, 19(16). doi:10.3390/ijerph191610274

See reference 13, 18 and 19

The experiments appear to be very dangerous (even with the safety equipment). The reviewer wonders how the study design was approved by the Ethics Committee. What are the standards?

The ethics committee considered the following:

- screening of participants ability to handle the motorcycle

- safety orientation of participants prior to the experiment

- driving experience of at least 5 years

- presence of a medical professional during the experiment

- supervision of a trained motorcycle professional during the experiment

- training of participants prior to the experiment

- safe environment to conduct the experiment

- screening of participants prior to the actual experiment and after the experiment.

Why only males were recruited for the two experiments? If the majority of motorcycle riders in Philippines are male, please provide the evidence.

We placed this as a limitation of the study.

Lines 425-427

Provide more information about the AIG. How are their replicability of alcohol-related vision impairment. BAC levels affect riding ability. How were the BAC levels considered?

More information about the AIG were included in section 2.1.2.

Lines 123-127

The effects of alcohol intoxication is not only vision but also motor skills, reaction time, etc. Please discuss.

This was discussed in paragraph preceding the conclusion.

Lines 417-421

Performance measures are not considered high. Are the models really capable to predict motorcycle rider’s intoxication?

Based on the experimentation, the model is able to achieved 100% accuracy from the training dataset. Also, if the training and test datasets were combined to train the model, it is able to achieve 100% accuracy. The performance is to show that future research should further investigate on how to further improve the  model with small dataset.

n/a

How can the developed model be useful? In the discussion, it was very briefly written. Include more policy implications.

This paragraph was added in the discussion section:

Governments will be able to use this technology to develop contactless monitoring of drunk MC drivers and influence manufacturers to integrate this feature in manufacturing safe motorcycles. Policies can be developed to empower manufacturers to make smarter MCs that can detect drunk driving and notify surrounding people or drivers of the driver's condition.

Lines 412-416

Round 3

Reviewer 3 Report

Thanks for your response.

However, I still believe the data collected on the motorcycle should be presented as is and not the body movement since it was not collected on the body directly. Moreover, the authors state it is based on visual observation of drivers in curves while their study is in straight lines. One would have collected data from the motorcycle simultaneously to the body to show that they can be considered the same.

Reviewer 4 Report

Although the authors improved the manuscript to some extent, the reviewer is not satisfied with it. Some answers (e.g., performance measures) are unable to understand the meaning. The largest concern about the mansucript was the usefulness and policy implications. The authors did not take the comment seriously about it.